# Stabilizing indium sulfide for $CO_2$ electroreduction to formate at high rate by zinc incorporation

Li-Ping Chi [1,2], Zhuang-Zhuang Niu [1,2], Xiao-Long Zhang [1,2], Peng-Peng Yang [1], Jie Liao [1], Fei-Yue Gao [1], Zhi-Zheng Wu [1], Kai-Bin Tang [1✉] & Min-Rui Gao [1✉]

Recently developed solid-state catalysts can mediate carbon dioxide ($CO_2$) electroreduction to valuable products at high rates and selectivities. However, under commercially relevant current densities of >200 milliamperes per square centimeter ($mA\ cm^{-2}$), catalysts often undergo particle agglomeration, active-phase change, and/or element dissolution, making the long-term operational stability a considerable challenge. Here we report an indium sulfide catalyst that is stabilized by adding zinc in the structure and shows dramatically improved stability. The obtained $ZnIn_2S_4$ catalyst can reduce $CO_2$ to formate with 99.3% Faradaic efficiency at $300\ mA\ cm^{-2}$ over 60 h of continuous operation without decay. By contrast, similarly synthesized indium sulfide without zinc participation deteriorates quickly under the same conditions. Combining experimental and theoretical studies, we unveil that the introduction of zinc largely enhances the covalency of In-S bonds, which "locks" sulfur—a catalytic site that can activate $H_2O$ to react with $CO_2$, yielding HCOO* intermediates—from being dissolved during high-rate electrolysis.

[1] Division of Nanomaterials & Chemistry, Hefei National Laboratory for Physical Sciences at the Microscale, University of Science and Technology of China, Hefei 230026, China. [2] These authors contributed equally: Li-Ping Chi, Zhuang-Zhuang Niu, and Xiao-Long Zhang. ✉email: kbtang@ustc.edu.cn; mgao@ustc.edu.cn

Electrosynthesis of value-added fuels using carbon dioxide ($CO_2$) as a feedstock provides an appealing route to reduce climate-changing $CO_2$ emission and a solid stepping-stone towards carbon neutrality[1,2]. Over the past decade, the development of catalysts that are active and selective for $CO_2$ reduction reaction ($CO_2$RR), and meanwhile, suppress the competing hydrogen evolution, has been the subject of intensive study. This has resulted in a variety of carbon-based products to be synthesized from $CO_2$, such as carbon monoxide (CO)[3], formate ($HCOO^-$)[4], methane[5], and higher hydrocarbons and oxygenates (e.g., ethylene[6], ethanol[7], and *n*-propanol[8]). Despite remarkable advances, recent techno-economic analyses showed that CO and formate are probably the only products that can achieve the industrialization trend of $CO_2$RR in the near future[9,10]. Regarding the formate, its profitable production requires current densities of $\geq 200$ mA cm$^{-2}$, Faradic efficiency (FE) of $>90\%$, and power conversion efficiencies of $>50\%$ (refs. [9,10]). Further, adopting solid-electrolyte electrolyzers permits the continuous production of formic acid without the separation process, making it even more economically viable[4].

Early research on $CO_2$RR from Hori and co-workers revealed that a number of metals, such as lead, mercury, indium (In), bismuth (Bi), cadmium, and tin, could convert $CO_2$ to formate, but many of these metals suffer from unsatisfactory selectivity or toxic issue[11]. Improvements in efficiency and selectivity have been achieved on nontoxic metallic catalysts via controlling of catalyst morphologies and dimensionalities[12,13], creation of vacancies[14], and introduction of other elements (e.g., O, S, and P) to form new phases[15–18]. When immobilized on the gas diffusion electrodes that surmount $CO_2$ mass transport limitation, commercially relevant rates ($>200$ mA cm$^{-2}$) and FE ($>90\%$) were observed to be reached on Bi nanosheets[19], $Bi_2O_3$ nanotubes[20], Bi metal-organic layers[21], and InP quantum dots[17]. However, the prospect on the potential of these catalysts for long-term operational stability is elusive. At high current densities, catalyst stability perhaps becomes a very important challenge. Often, $CO_2$RR activity deteriorates rapidly during high-rate electrolysis, owing to reasons like catalysis agglomeration[22], active-phase change[12,23], and element dissolution[15,24]. Unfortunately, previous research effort on catalyst stability, especially working at commercially relevant current densities, has remained rather rare. To make renewably powered formate electrosynthesis from $CO_2$ to be practical, it is critically necessary to develop catalysts that are not only active but also stable, and to gain insights on mechanisms of mediating the intrinsic stability.

Here, we report that incorporation of zinc (Zn) into indium sulfide ($In_2S_3$) synthesis enables tuning over its phase and structure, which dramatically improves the long-term stability of the resultant catalyst ($ZnIn_2S_4$) although the catalyst morphology remains almost unchanged. Comprehensive experiments coupled with computational studies reveal an enhanced covalency of In−S bonds mediated by Zn, which overstabilizes sulfur—a catalytic site that can activate $H_2O$ to react with $CO_2$, leading to the formation of HCOO* intermediates—in the catalyst structure. Consequently, we achieved nearly 100% $CO_2$-to-formate conversion at a current density of 300 mA cm$^{-2}$ over 60 h without degradation, corresponding to a high production rate of 8,894 μmol cm$^{-2}$ h$^{-1}$.

## Results

### Synthesis and characterizations of catalysts.
We had an interest in indium sulfide as a catalyst because S-doped In was shown by Wang and co-workers to be effective for catalyzing $CO_2$RR to formate. The presence of S enables facile activation of $H_2O$ to form adsorbed H*, which consequently reacts with absorbed $CO_2$ to yield HCOO* intermediates[16]. However, the stability of S-doped In was only assessed under ~60 mA cm$^{-2}$ during a 10 h period; the prospect of such catalyst for durable high-rate $CO_2$-to-formate conversion is unclear. Very recently, Xia et al. reported that exfoliated ultrathin $ZnIn_2S_4$ nanosheets with rich Zn vacancies show improved $CO_2$RR ability to formate[14]. Although interesting, its long-term stability at current densities relevant to commercial operation ($>200$ mA cm$^{-2}$) was not evaluated. These results motivated us to examine the ability of indium sulfide instead of S-doped In for mediating $CO_2$ to formate. We synthesized indium sulfide hydrothermally by the reaction of $InCl_3$·$4H_2O$ and $C_2H_5NS$ in deionized water (DIW) at 160 °C (Supplementary Fig. 1). Cubic $In_2S_3$ (JCPDS 65-0459; Fig. 1i) was produced after 6 h, exhibiting flower-like morphology composed of hierarchically organized nanosheets (Supplementary Fig. 2). Indeed, we observed good formate selectivity on $In_2S_3$, but the performance degraded quickly at high current densities owing to the dissolution of $S^{2-}$ ions (discussion later).

Previous experimental studies revealed that adding Zn in some transition metal chalcogenides (e.g., $Co_3S_4$)[25] can enhance the structure robustness. Thus we sought to improve the stability of high-rate $CO_2$RR by incorporating Zn into indium sulfide. We used the same hydrothermal method for preparing the desired product except the addition of $ZnCl_2$ during the synthesis (Supplementary Fig. 1). Intriguingly, we obtained hexagonal-structured $ZnIn_2S_4$ (JCPDS 65-2023; Fig. 1i) microflowers that consist of hierarchically organized nanosheets (Fig. 1a, b), which closely resemble $In_2S_3$ described above. The thicknesses of nanosheets were determined to be ~8.69 nm for $ZnIn_2S_4$ and 9.32 nm for $In_2S_3$ through atomic force microscopy (AFM) measurements (Supplementary Fig. 3). We note that the synthesis of flower-like $ZnIn_2S_4$ was previously reported[26–28], whereas the analogous morphologies of $ZnIn_2S_4$ and $In_2S_3$ that synthesized by the same protocol here will underpin a fair performance comparison. Energy-dispersive X-ray (EDX) spectrum elemental mapping exhibits a uniform spatial distribution of Zn, In, and S (Fig. 1c). This simple synthetic strategy enables the production of high-yield $ZnIn_2S_4$ material with good fidelity for potential large-scale adoption (Supplementary Fig. 4).

We studied the detailed atomic structure of the $ZnIn_2S_4$ by high-angle annular dark-field scanning transmission electron microscopy (HAADF-STEM). The atomic-resolution Z-contrast images in Fig. 1d, e clearly reveal hexagonal lattice, where In atoms exhibit higher image intensity than the overlapped Zn and S atoms (Supplementary Fig. 5). The fast Fourier transform (FFT) result exhibits the (100) and (010) reflections (Fig. 1f). Using image contrast, the In and Zn(S) atoms can be further identified by the line intensity profile (Fig. 1g) acquired along the yellow arrow in Fig. 1d. The corresponding atomic model depicts that all the overlapped Zn and S atoms are located at the centers of honeycomb (Fig. 1h). Without the addition of Zn leads to the crystallization of cubic $In_2S_3$ by the same synthetic protocol (Fig. 1i and Supplementary Fig. 2). Structurally, $ZnIn_2S_4$ belongs to $(ZnS)_mIn_2S_3$ ($m = 1$–3) system[29], which bears an orderly alternation of S and Zn(In) (Fig. 1b, right). The sequence of atoms along the [001] direction is S−$Zn_T$−S−$In_O$−S−$In_T$−S, where $Zn_T$ and $In_T$ occupy the tetrahedral coordination and $In_O$ occupies the octahedral site, respectively[29,30]. By comparison, one-third of the tetrahedral sites in $In_2S_3$ is unoccupied[31]. The incorporation of Zn alters the coordination environment of indium sulfide and thus might tailor favorably the electronic structure and catalytic properties.

To probe the electronic structures of $ZnIn_2S_4$ and $In_2S_3$, we measured the work function by ultraviolet photoelectron spectroscopy (UPS) (Fig. 1j). Our results show a lower work function of $ZnIn_2S_4$ (5.12 eV) compared to $In_2S_3$ (5.29 eV), revealing a superior electronic property by the incorporation of Zn element, consistent with electrochemical impedance spectroscopy (EIS) results (Supplementary Fig. 6). We speculate, on the basis of the

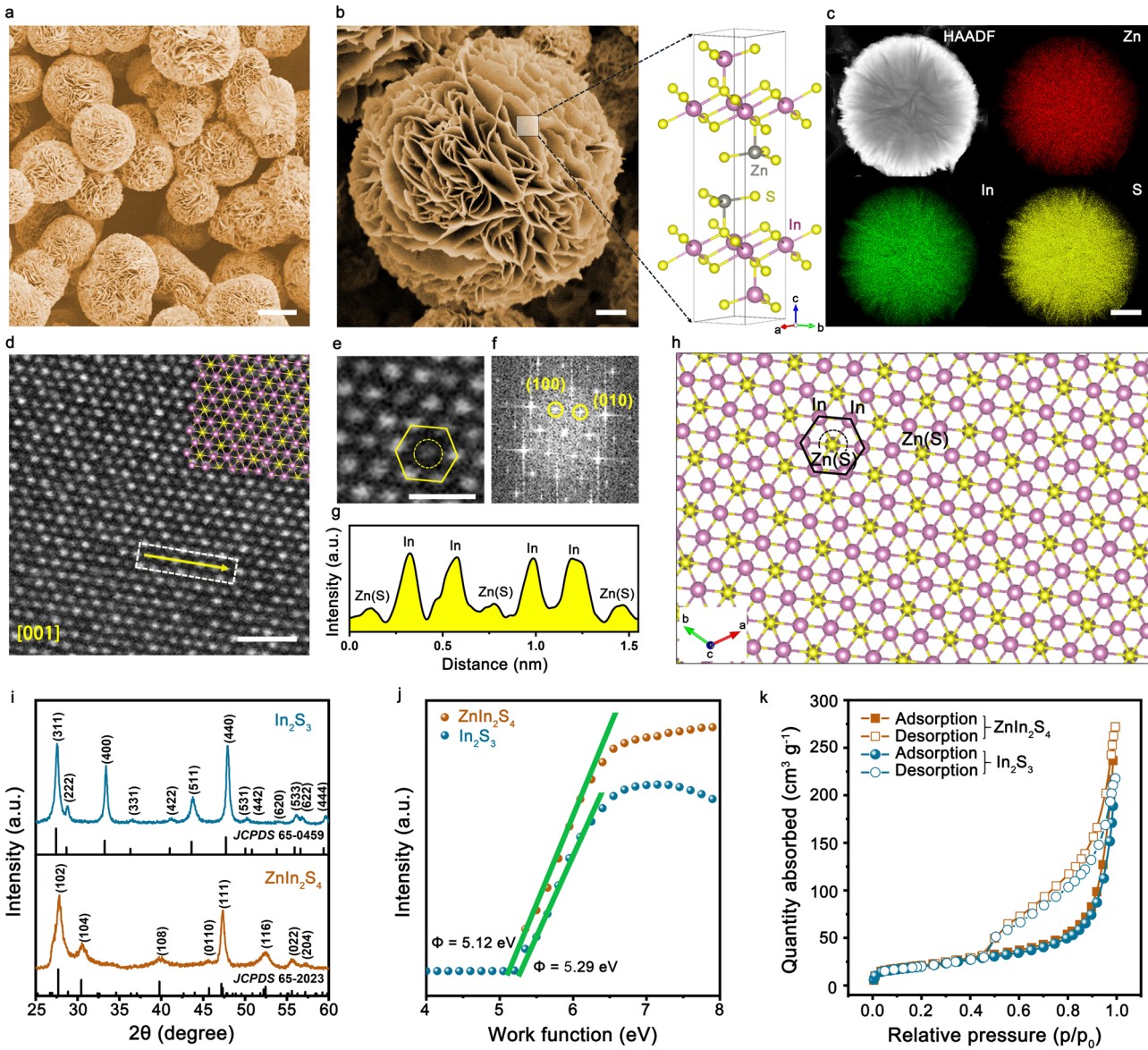

**Fig. 1 Physical characterization of ZnIn₂S₄. a, b** SEM images of the ZnIn₂S₄ catalyst. The right panel in **b** shows the crystal structure of ZnIn₂S₄. Scale bars, 5 μm (**a**) and 1 μm (**b**). **c** STEM-EDX elemental mapping of ZnIn₂S₄, exhibiting a uniform spatial distribution of Zn (red), In (green), and S (yellow), respectively. Scale bar, 1 μm. **d, e** Atomic-resolution Z-contrast images of ZnIn₂S₄ along [001] zone axis. Scale bars, 1 nm (**d**) and 0.5 nm (**e**). **f** The corresponding FFT pattern of (**d**). **g** The line intensity profile acquired along the yellow arrow in (**d**). **h** Atomic model of ZnIn₂S₄ along [001] zone axis. **i–k** XRD patterns (**i**), UPS spectra (**j**), and BET surface area analysis (**k**) of ZnIn₂S₄ and In₂S₃, respectively.

above results, that $CO_2RR$ may be highly favored on such ternary In-based sulfide owing to the modulated coordination environment and electronic structure. Moreover, we determined the Brunauer–Emmett–Teller (BET) surface areas of $ZnIn_2S_4$ and $In_2S_3$ to be 71.3 and 70.0 m³ g⁻¹ (Fig. 1k), respectively.

**CO₂RR performances in a flow cell**. We examined $CO_2RR$ properties of $ZnIn_2S_4$ and $In_2S_3$ catalysts in a flow cell (Supplementary Fig. 7) using recirculated 1 M $KHCO_3$ (pH 8.4) as electrolyte. $CO_2$ gas was fed at the cathode with a flow rate of 24 mL min⁻¹; the outlet gas flow rate was also measured for accurate product analysis (see "Methods"; Supplementary Fig. 8). We quantified the solution-phase and gas-phase products by using nuclear magnetic resonance (NMR) spectroscopy and on-line gas chromatography (Supplementary Fig. 9), respectively. The linear sweep voltammetry curves in Fig. 2a show sharp

reduction peaks for $ZnIn_2S_4$ and $In_2S_3$ catalysts in a $CO_2$ environment. In a $N_2$ environment, however, the two catalysts exhibit a slight current–voltage response. In comparison with $In_2S_3$, the onset potential for $CO_2RR$ on $ZnIn_2S_4$ catalyst shifted to a more positive value, implying enhanced $CO_2RR$ kinetics (Fig. 2a). Figure 2b shows that the Faradaic efficiency (FE) for formate on $ZnIn_2S_4$ catalyst was always greater than on $In_2S_3$ at all potentials examined (Supplementary Figs. 10, 11). Notably, the $ZnIn_2S_4$ catalyst yields peak FE of 99.3% for formate at −1.18 V versus a reversible hydrogen electrode (RHE), while the competing hydrogen evolution reaction (HER) on this catalyst was substantially suppressed (Fig. 2b and Supplementary Fig. 12). With this FE, we achieved a $CO_2RR$ to formate partial current density of ~298 mA cm⁻² (Fig. 2c), representing the highest value reported to date under $KHCO_3$ environments (Fig. 2e). We also performed reference measurements of hexagonal ZnS (JCPDS 39-

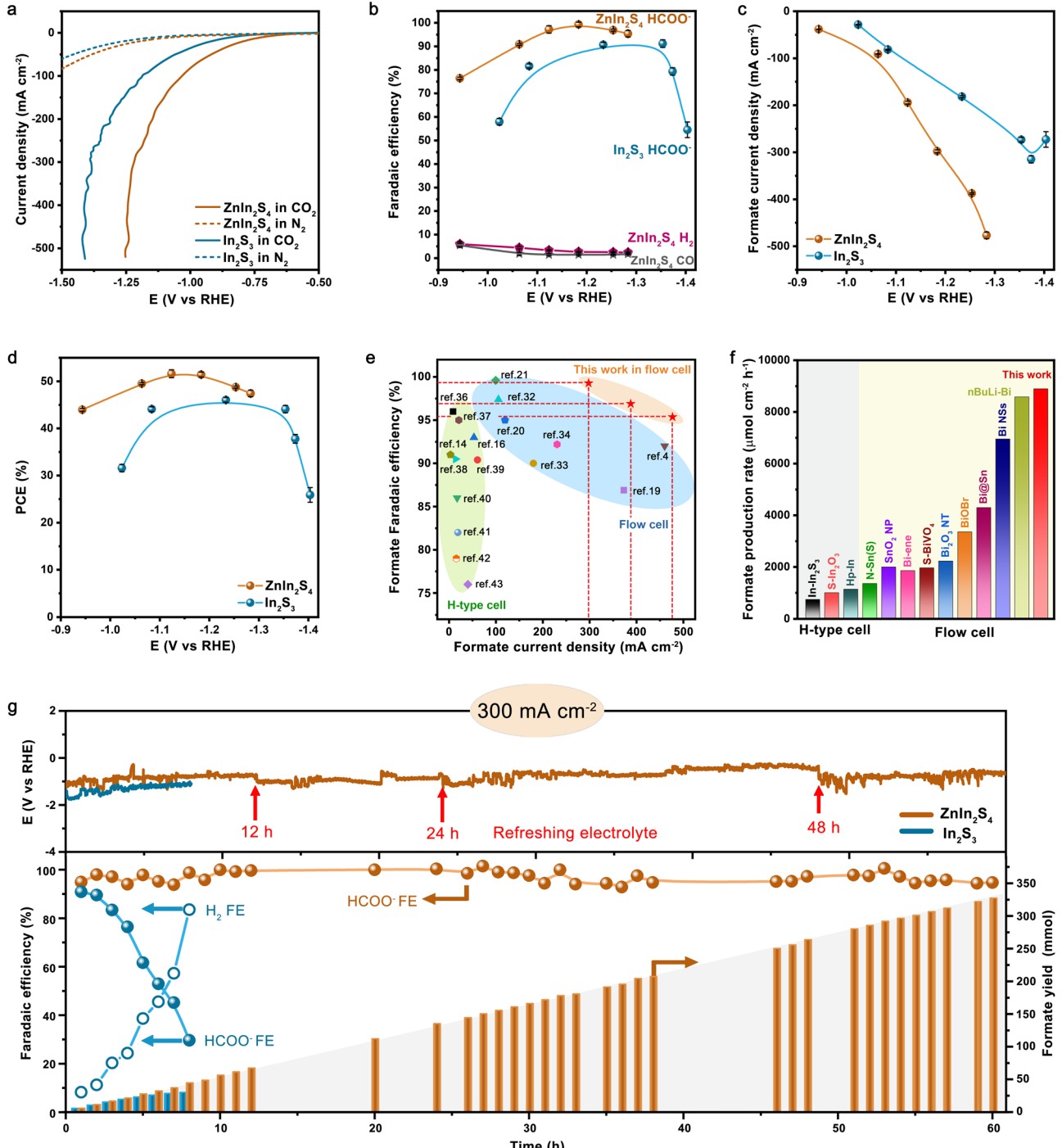

**Fig. 2 CO₂RR performances. a**, **b** The linear sweep voltammetry curves (**a**) and potential-dependent Faradaic efficiencies for products (**b**) on ZnIn₂S₄ and In₂S₃. **c**, **d** Partial current density (**c**) and half-cell PCE (**d**) for CO₂-to-formate conversion on ZnIn₂S₄ and In₂S₃. **e**, **f** Comparison of formate partial current densities and FEs (**e**), and formate production rates (**f**) for various catalysts reported under KHCO₃ environments (see Supplementary Table 1 for details). **g** Stability test of the ZnIn₂S₄ and In₂S₃ at 300 mA cm⁻². The electrolyte was occasionally replaced by new 1 M KHCO₃ solution (red arrows) to recover the ionic concentration and conductivity of the anolyte. The error bars represent the standard deviation of three independent measurements.

1363) that synthesized by the identical route for comparison, which, however, overwhelmingly produces H₂ (Supplementary Fig. 13). Additionally, our series of control experiments disclosed that the optimum CO₂RR performance was gained on ZnIn₂S₄ catalyst that hydrothermally synthesized at 160 °C for 6 h with a ZnCl₂:InCl₃·4H₂O ratio of 1:2 (Supplementary Figs. 14–19).

Figure 2d presents the half-cell power conversion efficiency (PCE) for CO₂-to-formate conversion under various applied

potentials. At −1.18 V versus RHE, our full-cell (CO₂ + H₂O → formate + O₂) device shows a half-cell formate PCE exceeding 50% on ZnIn₂S₄ catalyst. A comprehensive review of recent literature revealed that our ZnIn₂S₄ catalyst exhibits superb selectivity and partial current density (Fig. 2e), which result in a formate production rate of up to 8,894 μmol cm⁻² h⁻¹, outperforming all previous results[4,14,16,19–21,24,32–43] that have been reported under KHCO₃ environments (Fig. 2f).

We used density functional theory (DFT) to obtain insights into the $CO_2RR$ properties of the studied catalyst (see "Methods" for details). We compared the Gibbs free energies ($\triangle G$) for the formation of formate intermediate (HCOO*) on the surfaces of $ZnIn_2S_4$, $In_2S_3$, and In models (Supplementary Figs. 20–22). The computed barrier of HCOO* formation is 117 meV for $ZnIn_2S_4$ and 120 meV for $In_2S_3$, smaller than that of 270 meV for In, implying that S sites favorably mediate the HCOO* formation. Our calculations further reveal the lowest barrier of HCOOH* formation on $ZnIn_2S_4$, leading to its superior $CO_2$-to-formate ability. These results indicate that formate preferentially generates on $ZnIn_2S_4$ catalyst. By contrast, *COOH (intermediate of CO product) formation is highly endergonic on the three In-based catalysts (Supplementary Figs. 20–22), causing the production of CO to be virtually prohibited. Although HER process is largely hampered, our DFT results reveal that, as compared to $In_2S_3$ and In, the S sites of $ZnIn_2S_4$ enable much smaller hydrogen adsorption free energy of 370 meV. Early works[15,16] have reported that S acts as a promotor to enhance $CO_2$-to-formate conversion, we thus reasonably surmise that S sites on $ZnIn_2S_4$ surface permit easier $H_2O$ dissociation to adsorbed H* species, which then react with $CO_2$ to yield HCOO* intermediates.

**Comprehensive stability study**. Aside from activity, long-term stability—especially operating at high current densities ($> 200$ mA cm$^{-2}$)—is another critical metric for $CO_2$ electrolysis technique to be practical[10]. Figure 2g shows the key finding that we wish to report in this work: that is, the $CO_2RR$ stability of indium sulfide can be remarkably improved by the incorporation of Zn. We tested the stability of $ZnIn_2S_4$ catalyst at a profitable current density as large as 300 mA cm$^{-2}$, during this process portions of the electrolyte were frequently taken out for NMR analysis. The formate FE could be held at $> 97\%$ over 60 h of continuous electrolysis without the need of additional overpotentials (Fig. 2g). By contrast, $In_2S_3$ reference exhibited a rapid drop in formate selectivity, whereas the FE toward $H_2$ climbed up to ~90% within 8 h. We hypothesize that such severe performance drop might be caused by the structure degradation during high-rate electrolysis. Notably, the exceptional stability of $ZnIn_2S_4$ catalyst enables us to produce ~327 mmol formate after 60 h (Fig. 2g).

We combined multiple characterization techniques to track the structural evolution of $ZnIn_2S_4$ and $In_2S_3$ catalysts during $CO_2$ electrolysis under various current densities and operating times (Fig. 3). X-ray diffraction (XRD) patterns and scanning electron microscopy (SEM) studies reveal that the phase and morphology of $ZnIn_2S_4$ catalyst were well retained when progressively increasing the current density even up to 500 mA cm$^{-2}$ (Fig. 3a, c). By contrast, $In_2S_3$ catalyst undergoes a complete phase transition to metallic In (JCPDS 65-9292; Fig. 3b) at a current density of mere 50 mA cm$^{-2}$, accompanied by a dramatic morphology change (Fig. 3c, below images) owing to the loss of S that leads to structure collapse. Our Raman spectroscopy measurements on $ZnIn_2S_4$ show that two characteristic peaks at 248 ($LO_1$ mode) and 340 cm$^{-1}$ ($LO_2$ mode)[44,45] were retained after 60 h of operation at 300 mA cm$^{-2}$ (Fig. 3d). However, the characteristic Raman modes ($A_{1g}$ and $E_g$)[46] of $In_2S_3$ disappeared while Raman signals from metallic In (ref. [47]) were detected within mere 1 min (Fig. 3e and Supplementary Figs. 23a, 24). The Raman results are consistent well with our post-mortem SEM analyses (Supplementary Fig. 25) and XRD results (Supplementary Fig. 23b).

Of note that the severe loss of S for $In_2S_3$ catalyst was further verified by X-ray photoelectron spectroscopy (XPS; Fig. 3g), STEM-EDX elemental mapping (Fig. 3h) and SEM-EDX (Fig. 3i). This is starkly contrasted with $ZnIn_2S_4$ whose chemical state and content of each elements (i.e., Zn, In, and S) were nearly unaltered after 60 h of high-rate $CO_2$ electrolysis

(Fig. 3f, h and Supplementary Figs. 26, 27). We quantified the amount of S remained in $ZnIn_2S_4$ and $In_2S_3$ catalysts by using SEM-EDX (Supplementary Figs. 28–30), which permits a quantitative compositional analysis at a relatively large scale. As shown in Fig. 3i, the amount of S in $In_2S_3$ drops to 2.13 wt% from its original value (23.6 wt%) within the first 1 h, followed by a slow drop to almost zero over the next 2 h. This result is consistent with the significantly increased S amount in an electrolyte that measured by inductively coupled plasma atomic emission spectroscopy (ICP-AES; Supplementary Fig. 31). By contrast, the $ZnIn_2S_4$ catalyst shows negligible loss of S after 60 h. Moreover, selected-area electron diffraction (SAED) analyses of the used $ZnIn_2S_4$ catalyst reveal that the single-crystalline hexagonal phase well maintains after the aggressive long-term stability test (Fig. 3j). We further note that ZnS catalyst also performs very stable at high current densities (Supplementary Fig. 32) owing to the strong interaction between Zn and S (refs. [16,48,49]), although it mainly produces $H_2$ (Supplementary Fig. 13).

**Stability enhancement mechanism**. Our results above conclusively demonstrate that the stability degradation of $In_2S_3$ can be attributable to S leaching, and show the primacy of Zn as a stabilizer in indium sulfide that prevents S to be leached out. Besides indium sulfide, S leaching was also widely observed in other metal sulfides, while the dissolution mechanism is rather complex[6,18,50,51]. We turned to use DFT calculations to study the cause of the enhanced stability of indium sulfide after incorporating Zn. Compared with $In_2S_3$ having tetrahedral vacancies[31], the tetrahedral and octahedral sites in $ZnIn_2S_4$ are fully occupied[30] after Zn incorporation. Notably, in $ZnIn_2S_4$, all Zn atoms bind with S through tetrahedral coordination, which implies the formation of strong $Zn_T-S$ bonds considering that tetrahedral structures commonly give covalent feature[52]. Our computed differential charge density and its projection on the (110) plane map clearly reveal an enhanced electron cloud between Zn and S atoms (Fig. 4a, b), revealing electron donation from Zn to S due to the strong reducibility of Zn atoms. The transfer of electrons from In to S can also be seen more pronounced in $ZnIn_2S_4$ (Fig. 4a, b) than that in $In_2S_3$ (Fig. 4d, e), which leads to charge accumulation around In−S bonds and correspondingly higher covalency[53].

The calculated electronic localization function (ELF) of the tetrahedral $In_T-S$ and octahedral $In_O-S$ bonds in $ZnIn_2S_4$ are 0.84 and 0.79 (Fig. 4c), which compare larger than that of 0.71 and 0.76 in $In_2S_3$ (Fig. 4f), indicating a greater localization of $S-In_O-S-In_T-S$ (ref. [54]). Likewise, the ELF of tetrahedral $Zn_T-S$ bond in $ZnIn_2S_4$ was calculated to be 0.81 (Fig. 4c), pointing to its localized covalent feature. The interatomic bond strengths were further quantitatively analyzed by the projected crystal orbital Hamilton population (pCOHP) method (Fig. 4g−i). We found that the anti-bonding orbitals of In−S and Zn−S for $ZnIn_2S_4$ are less occupied. Moreover, our calculations yield integrated pCOHP values below the Fermi level of −0.763 and −0.737 for In−S bonds in $ZnIn_2S_4$ and $In_2S_3$, respectively, again demonstrating greater bond strengths in $ZnIn_2S_4$ (ref. [55]). These results, therefore, indicate that the bond breaking between In(Zn) and S in $ZnIn_2S_4$ is kinetically cumbersome, which explains the negligible S dissolution and thus exceptional long-term stability of the $ZnIn_2S_4$ catalyst (Supplementary Fig. 33).

## Discussion

In closing, we have shown long-term formate electrosynthesis from $CO_2$ at a high current density of 300 mA cm$^{-2}$ on a cost-effective

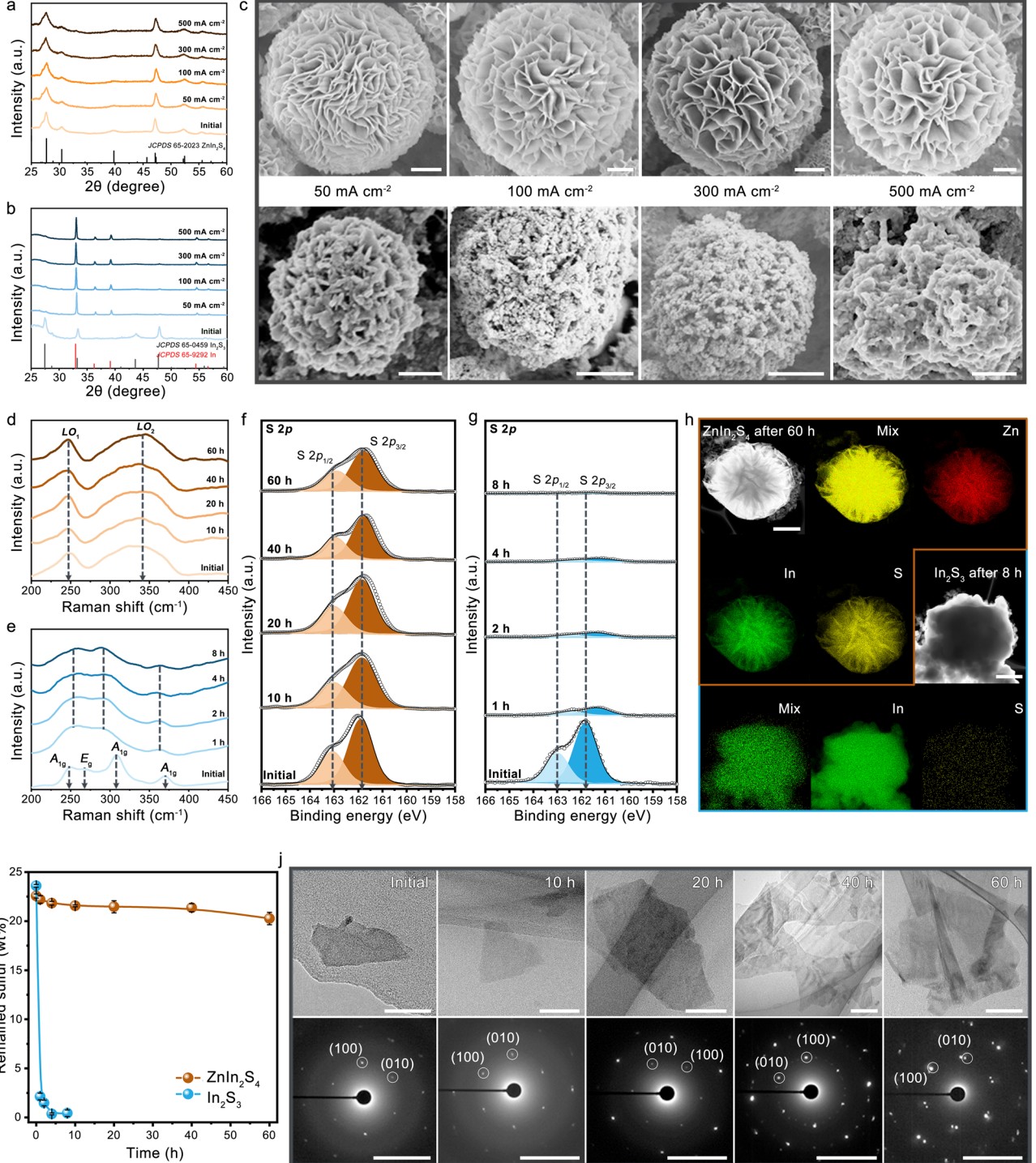

**Fig. 3 Structural stability of ZnIn₂S₄. a, b** XRD patterns of ZnIn₂S₄ (**a**) and In₂S₃ (**b**) after CO₂ /Subscript> electrolysis under various current densities for 10 min. **c** Corresponding SEM images of ZnIn₂S₄ (above) and In₂S₃ (bottom). Scale bars, 1 μm (above) and 500 nm (bottom). **d–g** Raman spectra of ZnIn₂S₄ (**d**) and In₂S₃ (**e**), and S 2p XPS spectra of ZnIn₂S₄ (**f**) and In₂S₃ (**g**) after CO₂ electrolysis for various times at 300 mA cm⁻². **h** STEM-EDX elemental mappings of ZnIn₂S₄ (scale bar: 1 μm) and In₂S₃ (scale bar: 600 nm) after running for 60 h and 8 h at 300 mA cm⁻², respectively. **i** SEM-EDX measurements of the remained sulfur in catalysts after running for various times at 300 mA cm⁻². The error bars represent the standard deviation of three independent measurements. **j** TEM (above, scale bars: 50 nm) and SAED patterns (down, scale bars: 5 1/nm) of ZnIn₂S₄ catalyst after CO₂ electrolysis for various times at 300 mA cm⁻².

indium sulfide catalyst modulated by Zn. The extraordinary catalyst stability can be explained by the increase of In−S covalency, which substantially prevents sulfur dissolution during CO₂RR. We achieved selective and fast CO₂-to-formate conversion with a formate FE of 99.3% and a notable formate production rate of 8,894 μmol cm⁻² h⁻¹. These findings will advance the development

of efficient and durable catalysts for commercial-scale electrosynthesis of formate.

## Methods

**Material synthesis**. All chemicals were used as received without further purification. Indium chloride tetrahydrate (InCl₃·4H₂O), thioacetamide (C₂H₅NS), and Zinc

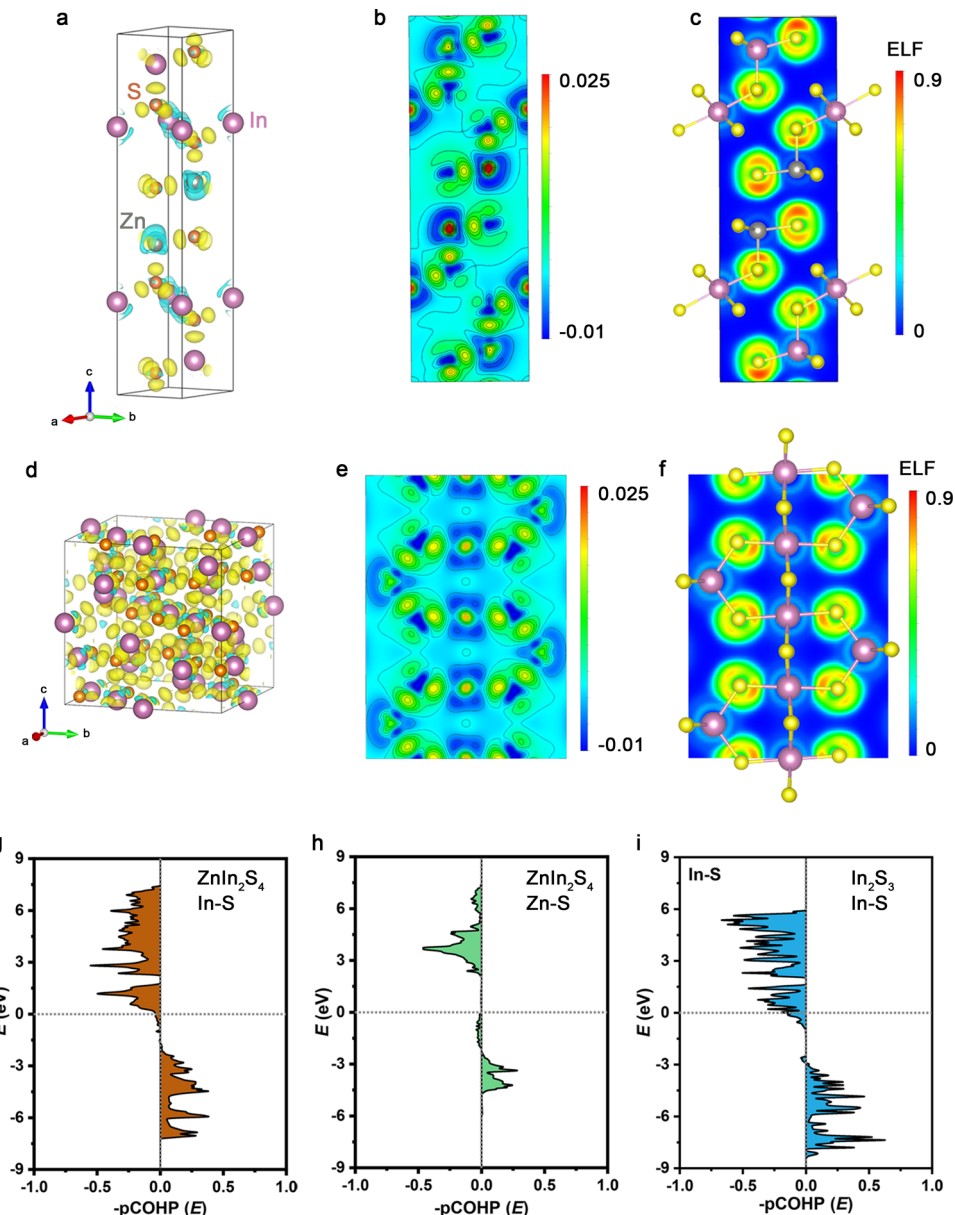

**Fig. 4 Enhanced covalency in ZnIn$_2$S$_4$. a**, **b** Differential charge density (**a**) and projection on the (110) plane (**b**). **c** ELF of ZnIn$_2$S$_4$. **d**, **e** Differential charge density (**d**) and projection on the (011) plane (**e**). **f** ELF of In$_2$S$_3$. The azure and yellow clouds represent electron density depressions and accumulations, respectively. **g–i** COHPs for In−S bonding (**g**) and Zn−S bonding (**h**) of ZnIn$_2$S$_4$, as well as In−S bonding (**i**) of In$_2$S$_3$.

dichloride (ZnCl$_2$), were purchased from Sinopharm Chemical Reagent Co., Ltd (Shanghai, China). In a typical experiment, 3.0 mmol InCl$_3$·4H$_2$O was dissolved in 150 mL deionized water (DIW), and then 6.1 mmol C$_2$H$_5$NS was added with vigorous stirring for 20 min. The 20 mL resultant solution was transferred into a 50 mL Teflon-lined autoclave, sealed, and heated at 160 °C for 6 h. After the reaction, the obtained In$_2$S$_3$ powders were washed with excess DIW and absolute ethanol for at least three times, and then dried at room temperature in an oven under vacuum for further characterization. The synthesis of ZnIn$_2$S$_4$ was the same with the synthesis of In$_2$S$_3$, except the addition of 1.5 mmol ZnCl$_2$ during the first step. For the synthesis of ZnS, it needs to replace 3.0 mmol InCl$_3$·4H$_2$O with 1.5 mmol ZnCl$_2$.

**Material characterizations**. XRD was performed on a Japan Rigaku DMax-γA X-ray diffractometer with Cu Kα radiation (λ = 1.54178 Å). The morphology of the samples was investigated by SEM (Zersss Supra 40) and TEM (JEOL 2010F(s)). The STEM and HRTEM images, SAED and EDX elemental mapping were taken on JEMARM 200 F Atomic Resolution Analytical Microscope with an acceleration voltage of 200 kV. SEM-EDX was determined by GeminiSEM 500 with an Oxford Aztec series X-ray energy spectrum. Raman spectra was measured on a Raman microscope (HORIBA) with a 785 nm excitation laser. ICP-AES data were obtained by an Optima 7300 DV instrument. N$_2$ adsorption/

desorption isotherms were recorded on an ASAP 2020 accelerated surface area and a porosimetry instrument (Mictromeritics), equipped with an automated surface area, at 77 K by using Barrett−Emmett−Teller calculations. XPS was taken on an X-ray photoelectron spectrometer (ESCALab MKII) with an X-ray source (Mg Kα $h\nu$ = 1253.6 eV).

**Preparation of CO$_2$RR electrodes**. The catalyst ink was prepared by ultrasonic dispersion of 10 mg catalyst powders in 1 ml isopropanol, which was mixed with 50 μL of 5 wt% Nafion. The resulted ink was uniformly spread on the gas diffusion layer (GDL, Sigracet 29 BC) of 3 × 3 cm$^2$ in the area by using an airbrush, yielding the prepared electrode with a catalyst loading of ~1.0 mg cm$^{-2}$.

**Electrochemical measurements**. All electrochemical measurements were performed in a flow cell with VSP-300 Potentiostat (Bio-Logic, France). For experiments in flow cells, gaseous CO$_2$ (99.999%) was passed through the gas chamber at the back side of the gas diffusion electrodes. Both catholyte and anolyte (1 M KHCO$_3$) were continuously circulated through the cathode and anode chambers separated by the cation exchange membrane (Nafion™ 117), which was used to avoid the crossover issues of formate[56]. The cathode is the prepared gas diffusion electrode (GDE, 1 × 1 cm$^2$), and the anode is a piece of nickel foam (1 × 1 cm$^2$).

The $CO_2$ inlet flowrates were kept constant at 24 mL min$^{-1}$ by a mass flow controller (C100L, Sierra). $KHCO_3$ electrolyte flowrates were maintained constant at 20 mL min$^{-1}$ controlled by a peristaltic pump (BT100-2J, Longer Pump). The $CO_2$ electrolysis lasted for 10 min unless otherwise specified. The linear sweep voltammetry (LSV) curves of $ZnIn_2S_4$ and $In_2S_3$ were performed in $CO_2$-fed and Ar-fed 1 M $KHCO_3$ solution. All potentials were measured against an Ag/AgCl (saturated KCl) reference electrode and converted to the RHE reference scale with $iR$ correction on account of the equation:

$$E\,(\text{vs RHE}) = E\,(\text{vs Ag/AgCl}) + 0.205 + (0.0591 \times \text{pH}) - iR_s \qquad (1)$$

Where the solution resistance $R_s$ was determined by EIS over a frequency range from 100 KHz to 10 mHz.

**$CO_2$RR products analysis**. The gas products were analyzed by gas chromatography (GC-2014, Shimadzu) equipped with thermal conductivity detector (TCD) to quantify $H_2$ concentration and flame ionization detector (FID) to analyze the content of CO. Considering $CO_2$ consumption, the outlet flow rate was monitored by a mass flowmeter (AST10-HLC, Asert Instruments) before flowing to the on-line GC. The Faradaic efficiency for gas products ($FE_x$) was calculated by the following formula:

$$FE_x(\%) = \frac{n_x \times C_x \times u \times F}{I \times V_M} \times 100\% \qquad (2)$$

where $F$ is the Faraday constant (96485 C mol$^{-1}$), $I$ is the total current density, $n_x$ is electrons transferred for reduction to product $x$, $C_x$ is volume fraction of the product $x$ detected by GC, $u$ is outlet gas flowrate and $V_M$ is molar volume (22.4 L mol$^{-1}$).

The formate products were quantified by $^1$H NMR spectra measured with a Bruker 400 MHz spectrometer. Typically, 400 μL of the electrolyte after $CO_2$RR electrolysis was mixed with 200 μL of $D_2O$ containing 50 ppm (m/m) dimethyl sulphoxide (DMSO) as the internal standard. The area ratio of the formate peak to the DMSO peak was compared to the standard curve to quantify the concentration of formate. The molar quantity of formate ($n_{formate}$) was calculated via multiplying the concentration of formate with the volume of the catholyte. The Faradaic efficiency of the formate ($FE_{formate}$) can be calculated by the following equation:

$$FE_{formate}(\%) = 2 \times F \times \frac{n_{formate}}{I \times t} \times 100\% \qquad (3)$$

where $t$ is the $CO_2$ electrolysis time.

The half cell (cathodic) power conversion efficiency (PCE, assuming the overpotential of the oxygen evolution reaction is zero) of the formate products was calculated using:

$$PCE(\%) = \frac{(1.23 - E_{formate}) \times FE_{formate}}{1.23 - E} \qquad (4)$$

where $E$ is the applied potential *vs* RHE, $E_{formate}$ is thermodynamic potential ($-0.02$ V vs RHE) of $CO_2$RR to formate[57].

The production rate for formate was calculated using the following equation:

$$\text{Production rate} = \frac{Q \times FE_{formate}}{F \times 2 \times t \times S} \qquad (5)$$

where $Q$ is the total charge passed and $S$ is the geometric area of the electrode (1 cm$^2$).

**DFT calculations**. The DFT calculations were performed by Vienna ab initio simulation package (VASP)[58] program with projector augmented wave (PAW)[59] method and the kinetic energy cut off was set to be 500 eV. The convergence criterion for electronic self-consistent iteration was set to be $10^{-4}$ eV. The atomic positions were fully relaxed until the force on each atom is less than 0.02 eV Å$^{-1}$. The Perdew−Burke−Ernzerhof[60] generalized gradient approximation exchange-correlation functional was used throughout. The slab model of In (101), $In_2S_3$ (311), and $ZnIn_2S_4$ (102) surface were constructed from the optimized In, $In_2S_3$, and $ZnIn_2S_4$ crystal structure. At the same time, a vacuum layer of 15 Å is established in the c-axis direction to ensure the separation between slabs. In addition, the surface formate species takes a unit negative charge, and, the present DFT calculation is not so great as to describe this kind of system carrying a neat charge. Thus, HCOOH was considered as the final product to describe this reaction rather than formate, in line with the DFT calculation in many researches[18,61,62]. The COHPs were computed using the developed lobster program[63–65].

Here, The Gibbs free energies were calculated at 25 °C and 1 atm:

$$\Delta G_{ads} = \Delta E_{ads} + \Delta ZPE - T\Delta S + eU \qquad (6)$$

where $\Delta E_{ads}$, $\Delta ZPE$, $T$, $\Delta S$, $U$, and $e$ are the binding energy, zero-point energies changes, temperature, entropy changes, applied potential at the electrode, and charge transferred, respectively.

## Data availability

All experimental data within the article and its Supplementary Information are available from the corresponding author upon reasonable request.

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

## Acknowledgements

This work is supported by the National Basic Research Program of China (Grant 2018YFA0702001), the National Natural Science Foundation of China (Grants 21975237 and 51702312), the Anhui Provincial Research and Development Program (Grant 202004a05020073), the USTC Research Funds of the Double First-Class Initiative (Grant YD2340002007), the Fundamental Research Funds for the Central Universities (WK2340000101), and the Recruitment Program of Global Youth Experts.

## Author contributions

M.R.G. and K.B.T. conceived and supervised the project. L.P.C. performed the experiments, collected and analyzed the data. Z.Z.N. helped with the test and analysis of NMR. X.L.Z. carried out the DFT calculations. P.P.Y., J.L., F.Y.G., and Z.Z.W. helped with electrochemical data collection and analysis. M.R.G., K.B.T., and L.P.C. co-wrote the paper. All authors discussed the results and commented on the paper.

## Competing interests

The authors declare no competing interests.
