## [Peer Review File · Nature Communications]

Title: Stabilizing indium sulfide for CO₂ electroreduction to formate at high rate by zinc incorporationREVIEWER COMMENTS

Reviewer #1 (Remarks to the Author):

This manuscript exhibits a thorough experimental investigation and analysis combined with computational works that neatly reveal the remarkable selectivity and stability of ZnIn₂S₄ catalyst for driving efficient CO₂-to-formate conversion at very high current densities (here 300 mA cm⁻²). The experiments were well done and the results are impressive. I particularly enjoy reading Figure 3 which clearly verifies that the morphology, phase, and composition remain nearly unchanged after such harsh high-rate electrolysis, which, however, degraded very fast without Zn incorporation. Such significant contrast highlights the effectiveness of the proposed strategy for improving catalyst stability. According to their studies, the authors attributed this stability improvement to the enhanced covalency of In-S bonds after introducing Zn, which is reasonable and scientifically sound.

I agree with the authors that maintaining high selectivity at commercially relevant current densities becomes a major challenge we are facing, which has to be surmounted before the CO₂ electrolysis technique to be practical. Thus, I do think this piece of work offers a good reference to the research topic of discovering new routes to enhance long-term stability of catalysts for real use.

Considering above reasons, I think this work is timely and deserves to be published in Nature Communications pending the below minor concerns are well clarified.

Minor comments:

---experiments in this manuscript undoubtedly show that S in In₂S₃ was almost completely dissolved in the electrolyte during the high-rate CO₂ electrolysis. Is it possible that these S ions would re-deposited on the electrode surface and affects the CO₂RR performances?

---authors also prepared ZnS as reference for CO₂RR studies, which produces mainly H₂ (Supplementary Fig. 11). Because Zn incorporation substantially improve the structural stability of indium sulfide, I wonder to see if S element is also “locked” in ZnS. Or does it readily dissolve during high-rate electrolysis? Results regarding this might be provided in Supplementary Information for readers’ information.

---indeed, the Zn incorporation can effectively stabilize indium sulfide for CO₂RR from experimental and simulation results. Does the amount of incorporated Zn affect the durability of the obtained catalysts?

Reviewer #2 (Remarks to the Author):

I have read with great interest the work on stabilizing indium sulfide for highly selective CO₂RR to formate at very large rate of 300 mA cm⁻² over 60 hours without performance loss. This was achieved

by incorporating cheap Zn in the structure which tailors favorably the electronic structure and local coordination environment. The authors combine experimental and computational studies to reveal much enhanced covalency of In-S bonds caused by Zn incorporation, which can protect sulfur (an active catalytic site that helps the formation of HCOO*) from being dissolved. The authors have put strong evidence on the fact that they indeed achieve remarkably robust CO₂RR catalyst and the results are mainly shown in Figure 3. This catalyst maintains high formate selectivity at commercially relevant current density of 300 mA cm⁻² over 60 hours of continuous electrolysis (mainly demonstrated in Figure 2g). Overall, I think this is a very beautiful work, which represents a solid step forwards to motivate CO₂ electrolysis technique to be practical. Given all of the above, I am strongly in favor of publication of this work pending several minor issues to be resolved.

1. The authors experimentally demonstrated the primacy of Zn as a stabilizer to prevent S to be leached out during high-rate CO₂RR. I am very curious about the structural stability of ZnS catalyst considering the existence of Zn. Would the Zn “lock” S in ZnS during CO₂ electrolysis?

2. In Supplementary Figure 21, operando Raman spectra of In₂S₃ show a quick transition from In₂S₃ to metallic In. But, Raman spectroscopy is an analytical technique for surface science. To study the structure change of In₂S₃, other characterization techniques that can probe the bulk information (e.g., XRD) needed to be carried out to solid the conclusion.

3. According to SEM images, the obtained catalyst has flower-like morphology composed of hierarchically organized thin nanosheets. The thickness of the nanosheets should be provided in the manuscript. Can the authors control the thickness of these sheets? Will the thickness affect the CO₂RR properties?

4. Details regarding the flow cell setups and the operando Raman setups should be provided in the Supplementary Information.

Reviewer #3 (Remarks to the Author):

There is no novelty to this manuscript whatsoever. The two main issues:

1. This work, or an otherwise eerily similar work, has already been published (<https://doi.org/10.1002/cssc.202002785>)

2. The authors claim novelty in the synthesis of ZnIn₂S₄. e.g. lines 91-96 "Previous experimental studies revealed that adding Zn in some transition metal chalcogenides 91 (e.g., Co₃S₄)₂₅ can enhance the structure robustness. Thus we sought to improve the stability of 92 high-rate CO₂RR by incorporating Zn

into indium sulfide. We used the same hydrothermal method 93 for preparing the desired product except the addition of $ZnCl_2$ during the synthesis (Supplementary 94 Fig. 1). Intriguingly, we obtained hexagonal-structured $ZnIn_2S_4$ (JCPDS 65-2023; Figure 1i) 95 microflowers that consist of hierarchically organized nanosheets (Fig. 1a and b), which closely 96 resemble In_2S_3 described above."

However, this is quite a well-known material for catalysis. The claim that the addition of Zn to the In_2S_3 structure to achieve $ZnIn_2S_4$ is surprising, much less a novel achievement that merits publication as a communication (in a high impact journal, no less), is an insult to the field of researchers who do their due diligence. It takes minutes with a search engine (e.g. google scholar) to find that the contents of this manuscript are not novel in any way. Furthermore, I find the discussion related to the synthesis of ZIS to be extremely ignorant of the extant literature in the field. ZIS syntheses were reported in detail in the 70s, and a large number of publications related to the synthesis of similar ZIS flower-like microspheres were reported in the late 00s and leading up to the present.

For the sake of scientific and academic integrity I would suggest the authors do their due diligence to make sure that their claims of novelty are, in fact, true.

I recommend rejection without the option for resubmission, and for Springer nature to overhaul their editorial "triage" process.

We highly appreciate all the reviewers for their valuable comments and suggestions that help us significantly improve the revised manuscript. We have carefully considered all the comments and revised our manuscript accordingly. Below is a point-by-point response to the reviewers' comments.

REVIEWER REPORTS:

Reviewer #1 (Remarks to the Author):

This manuscript exhibits a thorough experimental investigation and analysis combined with computational works that neatly reveal the remarkable selectivity and stability of ZnIn₂S₄ catalyst for driving efficient CO₂-to-formate conversion at very high current densities (here 300 mA cm⁻²). The experiments were well done and the results are impressive. I particularly enjoy reading Figure 3 which clearly verifies that the morphology, phase, and composition remain nearly unchanged after such harsh high-rate electrolysis, which, however, degraded very fast without Zn incorporation. Such significant contrast highlights the effectiveness of the proposed strategy for improving catalyst stability. According to their studies, the authors attributed this stability improvement to the enhanced covalency of In-S bonds after introducing Zn, which is reasonable and scientifically sound.

I agree with the authors that maintaining high selectivity at commercially relevant current densities becomes a major challenge we are facing, which has to be surmounted before the CO₂ electrolysis technique to be practical. Thus, I do think this piece of work offers a good reference to the research topic of discovering new routes to enhance long-term stability of catalysts for real use.

Considering above reasons, I think this work is timely and deserves to be published in Nature Communications pending the below minor concerns are well clarified.

Response: We thank the reviewer for the very positive assessment of our work and are grateful for his/her recommendation to publish the manuscript in Nature Communications.

1. Experiments in this manuscript undoubtedly show that S in In₂S₃ was almost completely dissolved in the electrolyte during the high-rate CO₂ electrolysis. Is it possible that these S ions would re-deposited on the electrode surface and affects the CO₂RR performances?

Response: We thank the reviewer for the thoughtful question. We combined SEM-EDX and XPS techniques to quantify the amount of S remained in In₂S₃ catalyst at 300 mA cm⁻². As shown in **Figure R1**, In₂S₃ catalyst undergoes severe loss of S with prolonging the electrolysis time. We did not observe any increased S signal during this period. Our ICP-AES data showed that S in the In₂S₃ catalyst almost completely dissolved in electrolyte (see **Supplementary Figure 31** in our revised Supplementary information). We note that, when a negative potential was applied, the S in the electrolyte exists in the form of S²⁻ or HS⁻ (*Metals*, 2016, 6, 23–53), which is hardly to redeposit on the electrode because of strong electrostatic repulsion.

Figure R1. Sulfur content measurements in In_2S_3 catalyst. SEM-EDX measurements of the remained sulfur in In_2S_3 catalyst (a, corresponding to **Supplementary Figure 28** in our revised Supplementary information) and S 2p XPS spectra (b, corresponding to **Figure 3g** in the revised manuscript) of In_2S_3 after CO_2 electrolysis for various times at 300 mA cm^{-2} .

2. Authors also prepared ZnS as reference for CO_2RR studies, which produces mainly H_2 (Supplementary Fig. 11). Because Zn incorporation substantially improve the structural stability of indium sulfide, I wonder to see if S element is also “locked” in ZnS. Or does it readily dissolve during high-rate electrolysis? Results regarding this might be provided in Supplementary Information for readers’ information.

Response: We thank the reviewer for the insightful question and good suggestion. Following your comments, we performed additional experiments. First, we used XRD to track the phase evolution of ZnS catalyst during CO_2 electrolysis under various current densities. Data in **Figure R2a** show that the phase of ZnS catalyst were also maintained well when the current density climbed up to 500 mA cm^{-2} . Further, we run the ZnS-modified electrode at 300 mA cm^{-2} for 8 hours and characterized the catalyst before and after use. Our SEM-EDX show that S maintained well in ZnS catalyst without obvious leaching (**Figure R2b**). Additionally, the morphology of ZnS was also well retained (**Figure R2c and d**).

Our results thus show that S in ZnS can be well maintained even at high current densities, which was explained by the strong interaction between Zn and S (see Ma, W. *et al.*, *Nat. Commun.* 2019, 10, 892; Wu, Y. *et al.*, *Nat. Commun.* 2021, 12, 3881; Hu, C. *et al.*, *Chem.* 2017, 3, 122-133). Although a very stable material, ZnS unfortunately is inactive for catalyzing CO_2RR electroreduction but favors

the competing hydrogen evolution reaction (see **Supplementary Figure 13** in our revised Supplementary information).

In the new version of our paper, we have added these data as **Supplementary Figure 32** and provided some discussions therein.

Figure R2. Stability evaluation of ZnS. **a**, XRD patterns of ZnS after CO₂ electrolysis under various current densities for 10 min. **b**, SEM-EDX of ZnS before and after 8 hours of operation at 300 mA cm⁻². Insets are the corresponding EDX selected areas. Scale bars, 1 μm. **c**, **d**, Corresponding SEM images of ZnS before (**c**) and after (**d**) CO₂ electrolysis for 8 hours at 300 mA cm⁻². Scale bars, 200 nm.

3. *Indeed, the Zn incorporation can effectively stabilize indium sulfide for CO₂RR from experimental and simulation results. Does the amount of incorporated Zn affect the durability of the obtained catalysts?*

Response: We thank the thoughtful question from the reviewer. Following your comments, we attempted to synthesize other ZnIn ternary sulfides with Zn:In ratios of 1:8, 1:4 and 1:1, which we denoted as ZnInS_x(1:8), ZnInS_x(1:4) and ZnInS_x(1:1), respectively. The total mole value of (Zn²⁺ + In³⁺) was kept at 4.5 mmol. Unfortunately, we can not gain pure phase by using these Zn:In ratios. As shown in **Figure R3a-d**, the XRD pattern of ZnInS_x(1:8) shows the formation of mixed In₂S₃ (JCPDS 65-0459) and ZnIn₂S₄ (JCPDS 65-2023) phases. When the Zn:In ratio was changed to 1:4,

the mixed diffraction peaks still exist, but with more ZnIn_2S_4 phase. As to $\text{ZnInS}_x(1:1)$, our XRD results indicate the coexistence of ZnIn_2S_4 and ZnS phases.

We also studied these mixed-phase catalysts at 300 mA cm^{-2} for 10 min. **Figure R3a-d** revealed that metallic In (JCPDS 65-9292) was emerged for both $\text{ZnInS}_x(1:8)$ and $\text{ZnInS}_x(1:4)$ because In_2S_3 exists, which readily loses S element. By contrast, $\text{ZnInS}_x(1:1)$ performs very stable at 300 mA cm^{-2} for 8 hours, yielding formate FE at $\sim 80\%$ without obvious decay (**Figure R3e**). Moreover, the amount of S in $\text{ZnInS}_x(1:1)$ and its morphology can be well maintained (**Figure R3f-h**). Although the inferior formate FE (the existence of ZnS would produce H_2 instead of formate), our results also imply that Zn incorporation is beneficial for the long-term stability of catalysts.

Figure R3. Structural stability of ZnInS_x . a-d, XRD patterns of $\text{ZnInS}_x(1:8)$ (a), $\text{ZnInS}_x(1:4)$ (b), $\text{ZnInS}_x(1:2)$ (c) and $\text{ZnInS}_x(1:1)$ (d) before and after CO_2 electrolysis for 10 min at 300 mA cm^{-2} . e, Stability test of the $\text{ZnInS}_x(1:1)$ at 300 mA cm^{-2} . f, SEM-EDX of $\text{ZnInS}_x(1:1)$ before and after 8 hours of operation at 300 mA cm^{-2} . Insets are the corresponding EDX selected areas. Scale bars, $1 \mu\text{m}$. g, h, Corresponding SEM images of $\text{ZnInS}_x(1:1)$ before (g, corresponding to **Supplementary**

Figure 18b in our revised Supplementary information) and after **(h)** CO₂ electrolysis for 8 hours at 300 mA cm⁻². Scale bars, 500 nm.

Reviewer #2 (Remarks to the Author):

I have read with great interest the work on stabilizing indium sulfide for highly selective CO₂RR to formate at very large rate of 300 mA cm⁻² over 60 hours without performance loss. This was achieved by incorporating cheap Zn in the structure which tailors favorably the electronic structure and local coordination environment. The authors combine experimental and computational studies to reveal much enhanced covalency of In-S bonds caused by Zn incorporation, which can protect sulfur (an active catalytic site that helps the formation of HCOO) from being dissolved. The authors have put strong evidence on the fact that they indeed achieve remarkably robust CO₂RR catalyst and the results are mainly shown in Figure 3. This catalyst maintains high formate selectivity at commercially relevant current density of 300 mA cm⁻² over 60 hours of continuous electrolysis (mainly demonstrated in Figure 2g). Overall, I think this is a very beautiful work, which represents a solid step forwards to motivate CO₂ electrolysis technique to be practical. Given all of the above, I am strongly in favor of publication of this work pending several minor issues to be resolved.*

Response: We are grateful for the reviewer's very positive assessment of our work.

1. The authors experimentally demonstrated the primacy of Zn as a stabilizer to prevent S to be leached out during high-rate CO₂RR. I am very curious about the structural stability of ZnS catalyst considering the existence of Zn. Would the Zn "lock" S in ZnS during CO₂ electrolysis?

Response: We thank the reviewer for the insightful question and good suggestion. Following your comments, we performed additional experiments. First, we used XRD to track the phase evolution of ZnS catalyst during CO₂ electrolysis under various current densities. Data in **Figure R4a** show that the phase of ZnS catalyst were also maintained well when the current density climbed up to 500 mA cm⁻². Further, we run the ZnS-modified electrode at 300 mA cm⁻² for 8 hours and characterized the catalyst before and after use. Our SEM-EDX show that S maintained well in ZnS catalyst without obvious leaching (**Figure R4b**). Additionally, the morphology of ZnS was also well retained (**Figure R4c and d**).

Our results thus show that S in ZnS can be well maintained even at high current densities, which was explained by the strong interaction between Zn and S (see Ma, W. *et al.*, *Nat. Commun.* 2019, 10, 892; Wu, Y. *et al.*, *Nat. Commun.* 2021, 12, 3881; Hu, C. *et al.* *Chem.* 2017, 3, 122-133). Although a very stable material, ZnS unfortunately is inactive for catalyzing CO₂RR electroreduction but favors the competing hydrogen evolution reaction (see **Supplementary Figure 13** in our revised Supplementary information).

In the new version of our paper, we have added these data as **Supplementary Figure 32** and provided some discussions therein.

Figure R4. Structural stability of ZnS. **a**, XRD patterns of ZnS after CO₂ electrolysis under various current densities for 10 min. **b**, SEM-EDX of ZnS before and after 8 hours of operation at 300 mA cm⁻². Insets are the corresponding EDX selected areas. Scale bars, 1 μm. **c**, **d**, Corresponding SEM images of ZnS before (**c**) and after (**d**) CO₂ electrolysis for 8 hours at 300 mA cm⁻². Scale bars, 200 nm.

2. In Supplementary Figure 21, *operando* Raman spectra of In₂S₃ show a quick transition from In₂S₃ to metallic In. But, Raman spectroscopy is an analytical technique for surface science. To study the structure change of In₂S₃, other characterization techniques that can probe the bulk information (e.g., XRD) needed to be carried out to solid the conclusion.

Response: We appreciate the useful comments from the reviewer. On the basis of your suggestion, we performed corresponding XRD studies on the In₂S₃ electrode after Raman measurements. As shown in **Figure R5b**, In₂S₃ catalyst undergoes a complete phase transition to metallic In within just one minute. We have added this new data as **Supplementary Fig. 23b** in the revised version of SI.

Figure R5. Operando Raman spectra and XRD patterns. The Operando Raman spectra (a, corresponding to **Supplementary Figure 23a** in the revised Supplementary information) and XRD patterns (b) of In_2S_3 under CO_2 electrolysis as a function of reaction time at 10 mA cm^{-2} in the operando Raman setups.

3. According to SEM images, the obtained catalyst has flower-like morphology composed of hierarchically organized thin nanosheets. The thickness of the nanosheets should be provided in the manuscript. Can the authors control the thickness of these sheets? Will the thickness affect the CO_2RR properties?

Response: We thank the reviewer for the good questions. To answer your question, we performed atomic force microscopy (AFM) measurements. From typical height profiles that show in **Figures R6a and b**, the nanosheet thickness were determined to be $\sim 8.686 \text{ nm}$ for ZnIn_2S_4 and 9.318 nm for In_2S_3 . Next, we tried to prepare ZnIn_2S_4 microflowers composed of nanosheets with different thicknesses. We found that the thickness of nanosheets can be increased simply by prolonging the hydrothermal reaction time, as shown in **Figure R6c-e**. We examined the CO_2RR performances of the three catalysts under various current densities in 1 M KHCO_3 , which indeed could effect the $\text{FE}_{\text{formate}}$ slightly, with the thickness of 8.686 nm showing the best performance. We have updated these new results as **Supplementary Figs. 14 and 15** in the revised version of SI.

Figure R6. Thickness measurement and CO₂RR performance. **a, b**, AFM images and corresponding height profiles of ZnIn₂S₄ (**a**) and In₂S₃ (**b**). Scale bars, 100 nm (**a**) and 200 nm (**b**). **c-d**, SEM images of ZnIn₂S₄ samples obtained at 160 °C for different reaction time: **c**, 6 h (corresponding to **Figure 1b** in the revised manuscript), **d**, 8 h (corresponding to **Supplementary Figure 14d** in the revised Supplementary information), **e**, 10 h, respectively. Scale bars, 1 μm. **f**, CO₂RR performance of the obtained samples (corresponding to **Supplementary Figure 15b** in the revised Supplementary information).

4. Details regarding the flow cell setups and the operando Raman setups should be provided in the Supplementary Information.

Response: We thank the reviewer for this good suggestion. The details regarding the flow cell setup (**Figure R7a and b**) and operando Raman setup (**Figure R7c and d**) were provided as **Supplementary Figs. 7 and 24** in the revised version of SI.

Figure R7. Photographs of the setups. a, b, Flow cell setups for CO₂RR measurements. **c, d,** The operando Raman setups in this work.

Reviewer #3 (Remarks to the Author):

There is no novelty to this manuscript whatsoever. The two main issues:

1. This work, or an otherwise eerily similar work, has already been published (<https://doi.org/10.1002/cssc.202002785>).

Response: We appreciate the reviewer's comments on our work. Indeed, the previous literature (<https://doi.org/10.1002/cssc.202002785>) mentioned by the reviewer reported the exfoliated ultrathin ZnIn₂S₄ nanosheets with rich Zn vacancies for catalyzing CO₂RR to formate. This is a very impressive study and perhaps is the only existed study that investigates ZnIn₂S₄ material for CO₂RR catalysis. This previous work has already been cited as **reference 14** in our original manuscript.

However, we should stress that the focus of this published paper and our current work was largely different. Now, we wish to state the difference of the two works from the following points:

In our work, **we, for the first time, uncovered that Zn in the ZnIn₂S₄ structure can greatly enhance the covalency of In-S bonds, which prevents the leaching of S during high-rate CO₂ electrolysis. Owing to such robust S sites, our catalyst can reduce CO₂ to nearly 100% formate at a commercially relevant current density of 300 mA cm⁻² over 60 hours of continuous operation without any performance loss.** This work makes a solid step toward industry-scale CO₂ electrolysis to make formate with profit.

By contrast, the published paper reported the creation of Zn vacancies in ultrathin ZnIn₂S₄ nanosheets by ultrasonic exfoliation. The generated Zn vacancies were found to reduce the energy barrier of CO₂ hydrogenation. **The long-term stability of this catalyst at current densities of >200 mA cm⁻²---current densities that require to enable profitable production of formate---was, however, not evaluated.**

Therefore, our work not only focused on activity and selectivity, **but, more importantly, aimed to resolve the long-term stability issue of catalysts at commercially relevant current densities, which currently represents a formidable challenge that needs to be overcome if we want the CO₂ electrolysis technique to be commercialized in the near future, as also echoed by the Reviewer 1# in his/her comments.**

In short, our study mainly focused on addressing the long-term stability issue of catalysts at high-rate CO₂ electrolysis, as well as understanding the enhancement mechanism behind. This obviously differs from the published paper mentioned by the reviewer.

We are sorry that we might not distinguish our work from this previous report clearly in our original manuscript, thus causing the confusion to the reviewer. To avoid misunderstanding, in the new version of our manuscript, the difference of the two works has been properly discussed.

2. The authors claim novelty in the synthesis of ZnIn₂S₄. e.g. lines 91-96 "Previous experimental studies revealed that adding Zn in some transition metal chalcogenides 91 (e.g., Co₃S₄)²⁵ can enhance the structure robustness. Thus we sought to improve the stability of 92 high-rate CO₂RR by incorporating Zn into indium sulfide. We used the same hydrothermal method 93 for preparing the desired product except the addition of ZnCl₂ during the synthesis (Supplementary 94 Fig. 1). Intriguingly, we obtained hexagonal-structured ZnIn₂S₄ (JCPDS 65-2023; Figure 1i) 95 microflowers that consist of hierarchically organized nanosheets (Fig. 1a and b), which closely 96 resemble In₂S₃ described above."

Response: We thank the reviewer for the comments, which, however, we can not agree with. We stress that, in our original manuscript, we **never claimed the novelty regarding the synthesis of ZnIn₂S₄**. The reviewer copied several sentences from our original manuscript to support his/her view. However, after reading these sentences, one can only feel that the authors just objectively state the obtained phase and morphology of the product, without any meaning that tones up the novelty of synthesis.

In fact, **our research goal here was not about synthesis. Instead, our important scientific finding here is that Zn incorporation in indium sulfide structure can "lock" S element from being dissolved owing to the substantially enhanced In-S covalency.** This feature thus enabled the resultant catalyst to reduce CO₂ to nearly 100% formate at a commercially relevant current density of 300 mA cm⁻² over 60 hours of continuous operation without any performance loss, **which was not achieved before.**

In our original manuscript, we indeed stated that this simple hydrothermal reaction allowed us to obtain catalysts with similar morphology and similar BET values. Similar morphology and BET values are actually **the basis of a fair performance comparison**, because potential influences from different morphologies and BET surface areas can be fully removed. **However, as we mentioned above, we never emphasized or toned up the novelty of synthesis in our paper. Again, we note that our work mainly focused on addressing the long-term stability issue of catalysts at high-rate CO₂ electrolysis (>200 mA cm⁻²), as well as understanding the mechanism behind.**

However, this is quite a well-known material for catalysis. The claim that the addition of Zn to the In₂S₃ structure to achieve ZnIn₂S₄ is surprising, much less a novel achievement that merits publication as a communication (in a high impact journal, no less), is an insult to the field of researchers who do their due diligence. It takes minutes with a search engine (e.g. google scholar) to find that the contents of this manuscript are not novel in any way. Furthermore, I find the discussion related to the synthesis of ZIS to be extremely ignorant of the extant literature in the field. ZIS syntheses were reported in detail in the 70s, and a large number of publications related to the synthesis of similar ZIS flower-like microspheres were reported in the late 00s and leading up to the present.

Response: Thanks for the comments. Still, the reviewer gave his/her comments solely on the synthesis and on the material itself. Unfortunately, these comments seem very conservative, lacking the confidence of exploring new science, new phenomenon, and new applications for an existing material.

First, we fully agree with the reviewer that “**this is quite a well-known material for catalysis**”. However, **to our best knowledge, all the previous ZnIn₂S₄ materials were studied in the photocatalysis field.** Works of using ZnIn₂S₄ as CO₂RR electrocatalysts are very rare. Perhaps the paper (<https://doi.org/10.1002/cssc.202002785>) mentioned by the reviewer **was the first study of evaluating the CO₂RR performance of ZnIn₂S₄.** Our work probably is the second one, which demonstrates the promise of using this catalyst for long-term CO₂ electrolysis to produce formate with nearly 100% selectivity at commercially relevant current density.

Second, the reviewer commented that “It takes minutes with a search engine (e.g. google scholar) to find that the contents of this manuscript are not novel in any way”. Unfortunately, we can not agree with this comment. As discussed above, in this work, **we, for the first time, uncovered that Zn in the structure can greatly enhance the covalency of In-S bonds, which prevents the leaching of S during high-rate CO₂ electrolysis.** Owing to such robust S sites, the resultant catalyst can **reduce CO₂ to nearly 100% formate at a commercially relevant current density of 300 mA cm⁻² over 60 hours of continuous operation without any performance loss.** These are new findings. **We are confident that these important results demonstrated in our paper have never been reported thus far.**

Third, we accepted the comments from the reviewer that the synthesis of ZIS was not comprehensively discussed in our original paper. Following your comments, we offered more discussions on the synthesis of this material in our revised version of paper.

Overall, this reviewer mainly commented on the synthesis of this material and totally ignored the key scientific findings we reported in this work. These comments also omitted the great potential of developing ZnIn₂S₄ for new applications, such as long-term and high-rate CO₂ electrolysis. These comments we think, on the basis of above considerations, are improper and are unfair to us.

For the sake of scientific and academic integrity I would suggest the authors do their due diligence to make sure that their claims of novelty are, in fact, true.

I recommend rejection without the option for resubmission, and for Springer nature to overhaul their editorial "triage" process.

Response: One more time, we thank the reviewer for taking valuable time to prepare these comments, although which we think are improper and are unfair to us. We guess that this reviewer is an expert on material synthesis, thus completely omitting the key scientific findings we reported in this paper. We sincerely hope that the reviewer will reevaluate this work after reading our responses and our revised paper carefully.

REVIEWERS' COMMENTS

Reviewer #1 (Remarks to the Author):

The authors have now answered my questions listed in the first round and I believe the manuscript now is ready for publication.

Reviewer #2 (Remarks to the Author):

The revision has addressed early comments. It can be accepted as it is.

Reviewer #4 (Remarks to the Author):

The authors report that notable results for CO₂RR performance by using ZnIn₂S₄ catalyst. They found that the stability of In-S can be significantly improved by Zn incorporation. Also, it should be noted that stabilization of In-S to guarantee highly-selective CO₂RR at large rate of 300 mA cm⁻² over 60 hours without performance loss.

There were some issues both in experimental and theoretical considerations, which are pointed out by reviewers in the first review process, in the original manuscript, but most of them have been properly addressed in the revised manuscript.

Regarding the originality of the research, as the authors claimed in the response letter, approach to enhance the In-S covalency by Zn incorporation is considered to be different from previous studies. Thus, I recommend publication of this paper in Nature Communications in its current form without any further revision.

P. S. The point-by-point answers to the referees' comments

REVIEWERS' COMMENTS:

Reviewer #1 (Remarks to the Author):

The authors have now answered my questions listed in the first round and I believe the manuscript now is ready for publication.

Response: We thank the reviewer for strong support on the publication of this work.

Reviewer #2 (Remarks to the Author):

The revision has addressed early comments. It can be accepted as it is.

Response: We thank the reviewer for strong support on the publication of this work.

Reviewer #4 (Remarks to the Author):

The authors report that notable results for CO₂RR performance by using ZnIn₂S₄ catalyst. They found that the stability of In-S can be significantly improved by Zn incorporation. Also, it should be noted that stabilization of In-S to guarantee highly-selective CO₂RR at large rate of 300 mA cm⁻² over 60 hours without performance loss.

There were some issues both in experimental and theoretical considerations, which are pointed out by reviewers in the first review process, in the original manuscript, but most of them have been properly addressed in the revised manuscript.

Regarding the originality of the research, as the authors claimed in the response letter, approach to enhance the In-S covalency by Zn incorporation is considered to be different from previous studies.

Thus, I recommend publication of this paper in Nature Communications in its current form without any further revision.

Response: We thank the reviewer for strong support on the publication of this work.